# Acute sore throat and *Fusobacterium necrophorum* in primary healthcare: a systematic review and meta-analysis

Stefan Malmberg [ID],[1,2] Susanna Petrén,[1] Ronny Gunnarsson,[1,2,3] Katarina Hedin,[4,5,6] Pär-Daniel Sundvall[1,2]

For numbered affiliations see end of article.

**Correspondence to**
Dr Stefan Malmberg;
stefan.malmberg@stfn.se

## ABSTRACT

**Purpose** The main objective of this review was to describe and quantify the association between *Fusobacterium necrophorum* (FN) and acute sore throat in primary healthcare (PHC).

**Methods** In this systematic review and meta-analysis, we searched Scopus and PubMed for case–control studies reporting the prevalence of FN in patients attending primary care for an uncomplicated acute sore throat as well as in healthy controls. Only studies published in English were considered. Publications were not included if they were case studies, or if they included patients prescribed antibiotics before the throat swab, patients with a concurrent malignant disease, on immunosuppression, having an HIV infection, or patients having another acute infection in addition to a sore throat. Inclusion criteria and methods were specified in advance and published in PROSPERO. The primary outcome was positive etiologic predictive value (P-EPV), quantifying the probability for an association between acute sore throat and findings of FN in the pharynx. For comparison, our secondary outcome was the corresponding P-EPV for group A *Streptococcus* (GAS).

**Results** PubMed and Scopus yielded 258 and 232 studies, respectively. Removing duplicates and screening the abstracts resulted in 53 studies subsequently read in full text. For the four studies of medium to high quality included in the meta-analysis, the cumulative P-EPV regarding FN was 64% (95% CI 33% to 83%). GAS, based on data from the same publications and patients, yielded a positive EPV of 93% (95% CI 83% to 99%).

**Conclusions** The results indicate that FN may play a role in PHC patients with an acute sore throat, but the association is much weaker compared with GAS.

## INTRODUCTION

An uncomplicated acute sore throat is a common reason for attending primary healthcare (PHC).[1–3] Most current guidelines concerning the management of patients with a sore throat focus on group A *Streptococcus* (GAS).[4–6] However, recent studies have indicated that *Fusobacterium necrophorum* (FN) might cause a sore throat, particularly among adolescents and young adults.[7–9]

FN is an anaerobic Gram-negative bacterium most known for causing the severe

### Strengths and limitations of this study

► This is the first systematic literature review with meta-analysis using positive etiologic predictive value to quantify the clinical relevance of a finding of *Fusobacterium necrophorum* (FN) in patients presenting with an uncomplicated acute sore throat in primary healthcare (PHC).
► The positive etiologic predictive value reveals the probability for a true link between FN and the sore throat expressed as a plain percentage between 0% and 100% and it was 64% (95% CI 33% to 83%).
► A potential limitation is that there were only four available case–control studies with low or medium risk for bias presenting the proportion of FN.

disease Lemierre's syndrome, a potentially life-threatening condition that typically begins with a sore throat and is also an established pathogen in peritonsillar abscess (PTA).[10–12]

The role of FN in the sore throat has been studied in three recent reviews.[7–9] None of the three reviews have taken into consideration the carriage rate of FN in healthy controls, which is of importance when estimating the clinical relevance of finding FN in patients with an uncomplicated acute sore throat.

This study aimed to estimate the probability for an association between FN and the uncomplicated acute sore throat in patients attending PHC, when taking into consideration the carriage rate of FN in healthy controls. A second aim was to compare the probability for FN with the same probability for an association between GAS and patients with an uncomplicated acute sore throat.

### Patient and public involvement

No patients involved.

### Ethics approval

No ethical approval was needed since only publicly available data from published articles (in which informed consent was obtained by



the primary investigators) were retrieved and analysed. No personal, sensitive or confidential information was collected in the scope of this study.

## METHODS
The review was conducted in accordance with the Preferred Reporting Items for Systematic Reviews and Meta-Analyses (PRISMA) statement.[13] Inclusion criteria and methods were specified in advance and documented in the review protocol. The initial protocol was registered and made available beforehand in PROSPERO (International Prospective Register of Systematic Reviews), 5 September 2018 (registration number CRD42018106800).

### Search strategy
PubMed and SCOPUS were searched (17 September 2018) for case–control studies reporting the prevalence of FN in patients with an uncomplicated acute sore throat and in healthy individuals without any signs of infection. There were no time limitations. The search terms are described in online supplemental appendix 1.

### Study selection
All case–control studies reporting the prevalence of FN in patients attending a PHC setting for an uncomplicated acute sore throat and in a healthy control group were included. Only studies published in English were considered. Publications were not included if they were case studies, or if they included patients prescribed antibiotics before the throat swab, patients with a concurrent malignant disease, on immunosuppression, having an HIV infection, or patients having another acute infection in addition to a sore throat.

SM performed the first screening by reading titles and abstract to remove duplicates from the two search strategies and, thereafter, to remove obviously irrelevant studies such as animal studies. The remaining studies were carefully screened again reading titles and abstracts independently by SM and SP to identify studies that potentially met the inclusion criteria outlined above. SM and SP started screening sitting together in the same room discussing each publication to ensure they aligned their judgement. They then continued screening separately but had a joint discussion whenever they decided differently if a publication should be kept or removed

The full texts of these potentially eligible studies were retrieved and independently assessed for eligibility by SM and SP. Any disagreement between them over the eligibility of particular studies was resolved through discussion within the whole review team.

The reference lists for each article were screened for additional articles potentially matching the inclusion criteria. Such articles were added to the list of potentially eligible studies for further assessment.

### Appraisal of methodological quality
SM and SP independently assessed the risk of bias in the included studies by using methodological quality characteristics (table 1). Overall high quality was defined as having a low risk of bias in all criteria. Having a high risk of bias in any criteria made the study to be of overall low quality. The rest was classified as having an overall moderate risk of bias. Disagreements over the risk of bias in particular studies were resolved by discussion within the review team.

### Data extraction
A standardised, pre-piloted form was used to extract data from the included studies for assessment of methodological quality and evidence synthesis. Extracted information included study setting, definition of cases and classification of these using the Centor criteria if available,

**Table 1** Methodological quality assessment of included studies

| | Low risk | Intermediate risk | High risk |
|---|---|---|---|
| Definition of cases | Cases well defined as per Centor criteria or similar | At least two criteria mentioned in case definition | Cases not defined |
| Healthy controls | Study includes comparison with asymptomatic controls | Controls not asymptomatic | – |
| Swab method | Area of throat swabbed described, transport and storage mentioned | Area of throat swabbed mentioned but not the transport or storage | No mention of swab method |
| Culture method | Clear description of culture media, incubation time (or PCR if used) | Method described but not in detail | Method not discussed |
| Type of study | Case–control studies on FN | Community surveillance studies mentioning FN prevalence | Observational studies without well-defined cases and controls |
| Same area and time period | Cases and controls are collected in the same area and time of year | Cases and controls are collected in the same area but over different time periods | Cases and controls are collected in different regions and time periods |

FN, *Fusobacterium necrophorum*.

definition of healthy controls, swab method, culture or PCR method, outcomes of throat swabs and information for the assessment of the risk of bias. SM and SP extracted data independently. Discrepancies were identified and resolved through discussion.

## Data analysis

A narrative synthesis was produced for each of the included studies, structured around the study methodology, target population characteristics, outcome and the assessment of methodological quality.

Studies with a healthy control group and of a medium to high methodological quality were used for the meta-analysis, where the pooled difference in prevalence of FN between cases and healthy controls was compared using $\chi^2$ test.

The clinical relevance of any statistical differences between symptomatic patients and healthy controls was further explored by calculating the positive etiologic predictive value (P-EPV).[14] P-EPV is a method of quantifying the probability of a true link between the symptom (a sore throat) and the finding of FN in the throat while considering the possibility of healthy carriers of FN (online supplemental appendix 2). P-EPV for FN was, when possible, compared with P-EPV for GAS using data from the same patients and publications.

Using a random effects meta-analysis would have provided ORs for harbouring FN among cases compared with controls. The statistical technique we used for meta-analysis, P-EPV, has been used previously[15] and provides a probability for a true connection between FN and the symptom of a sore throat in the studied group. This outcome is a plain percent between 0% and 100%. If a bacterium is found equally often in patients and controls, the point estimate of P-EPV will be 0% with a 95% CI from 0.0% and the upper limit will be determined by the sample size. The point estimate of P-EPV will approach 100% when the difference in prevalence of a bacterium between patients and controls increases. This was in our opinion a more clinically useful outcome that can be easily understood, especially by clinicians unfamiliar with research and ORs.

P-EPV does not directly take into account between-study variation so it is not a random effect model. We compensate for this by presenting our outcome for each individual study as well as a sensitivity analysis where we compare the consequences of combining them differently. Furthermore, the between-study variation statistics ($I^2$) calculated in random effect models is very unreliable when the number of included publications is small[16] and we knew already from the start that the number of available publications would be small.

P-EPV has the ability to adjust the proportion of individuals harbouring FN between healthy controls and symptomatic carriers ill from something else like a virus. However, in this review we chose to not adjust this, so we set theta to 1.0, meaning that symptomatic carriers harbour FN equally often as healthy controls. Finally,

*Acute Sore Throat and Fusobacterium Necrophorum in PHC – A Systematic Review and Meta-analysis*
Version: (15/7/20 12:12:00 pm)

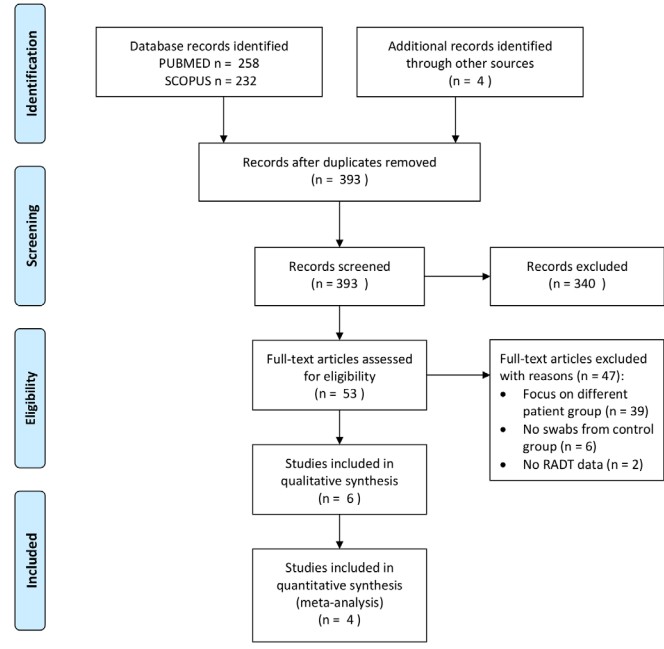

PRISMA 2009 Flow Diagram

From: Moher D, Liberati A, Tetzlaff J, Altman DG, The PRISMA Group (2009). *Preferred Reporting Items for Systematic Reviews and Meta-Analyses: The PRISMA Statement.* PLoS Med 6(7): e1000097. doi:10.1371/journal.pmed1000097

For more information, visit www.prisma-statement.org.

**Figure 1**   PRISMA (Preferred Reporting Items for Systematic Reviews and Meta-Analyses) flow diagram. RADT, rapid antigen detection test.

P-EPV allows us to consider the sensitivity of the test to detect FN, something that conventional random effects meta-analysis does not.

## RESULTS

The PubMed search yielded 258 publications, and the Scopus database query yielded 232 (figure 1). Reviewing reference lists did not reveal any more relevant publications not found in the initial searches. Removing duplicates and screening the abstracts resulted in 53 studies subsequently read in full text.

### Exclusion of publications

Thirty-seven of these 53 articles were not included because they had a different focus, that is, laboratory methods,[17] or focused on a different category of patients than was the scope of this review: chronically ill patients,[18] hospitalised patients,[19–24] or patients with a subset of infections such as PTA, Lemierre's syndrome, chronic/recurrent tonsillitis and intra-abdominal infections.[11 25–52] Four were excluded because they lacked a control group.[53–56] Discussions in the review team prompted the exclusion of another five articles with methodological limitations in relation to the scope of this review.[57–61]

The article published in 2004 by Aliyu et al[57] concerned a study where routine throat swabs were analysed for FN-specific DNA and compared with swabs obtained from healthy adults. The cases were randomly selected in the laboratory, but it was unclear to what extent they had a sore throat and how these symptoms were registered. The cases included children as young as 5 months. Hence, it was unclear what kind of patients the routine throat swabs sent to the laboratory represented. Inclusion in the control group required the absence of antibiotic therapy in the preceding 2 weeks, but not for cases. The swabs from cases were also cultured for GAS, but there was no information about the prevalence of GAS in the control group. The mean age and range differed substantially between cases and controls.

The article published in 2018 by Atkinson et al[58] presented the results of applying a new laboratory method on swabs from a previously published study[62] and comparing the results to the method that was used initially. The cases included university students complaining of a sore throat, while the controls were asymptomatic students. Only 30 of the 180 control swabs were used in this study, limiting the validity of the comparison between the different laboratory methods. This limitation is highlighted by the remarkable difference in outcome between the laboratory methods when reviewing cases and controls for both FN and GAS.

A Letter to the Editor published in 2014 by Eaton et al[59] concerned a project in which all throat swabs received by a microbiology laboratory in 1 year were cultured for GAS and FN, indicating that the only inclusion criterion was that a throat swab was taken. The laboratory served both PHC and secondary care, while the scope of this study was to focus on uncomplicated acute sore throat in PHC. Clinical details stated on accompanying request forms were used to determine if patients had pharyngitis. Those with either persistent or recurrent symptoms were considered to have persistent sore throat syndrome (PSTS), indicating that multiple swabs from the same individual were allowed in the data. There was no information about antibiotic treatment. In conclusion, it was decided not to include the Letter to the Editor by Eaton et al[59] in this review.

The text published in 2009 by Ludlam et al[60] described a study design comparing cases from a local general practitioner to controls comprising healthy university students, collecting throat swabs for both groups during the same 2-month period. The description of inclusion and exclusion criteria for the cases was limited. The results for GAS, Epstein-Barr virus and FN were not shown for both groups, either in text or in tables. It is unclear whether the controls may become cases (and vice versa). Their online supplemental table contained information about antibiotic treatment in the control group, but not for the cases.

The article published in 2018 by Pallon et al[61] was a follow-up study based on the same initial data already included in a previously published article by Hedin et al,[63]

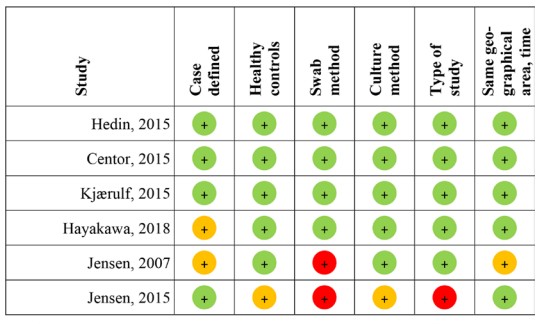

**Figure 2** Quality assessment of included studies.

which is included in this review. Therefore, the article published in 2018 by Pallon et al[61] was not included.

## Methodological quality

Of the six studies included in the qualitative analysis, three were of overall high quality and one of medium quality[62–65] (figure 2), and these were included in the meta-analysis.

Two studies presenting data from cases and controls were of low quality,[66 67] and these were not included in the meta-analysis for the reasons described below.

The Danish study by Jensen et al 2015[67] examined the outcome of throat cultures arriving at a microbiology laboratory. Most of the cases came from patients with 3–4 Centor criteria who had already tested negative with a rapid antigen detection test (RADT) for GAS. The control group consisted of subjects having a sore throat with 0–2 Centor criteria or fever or who were screened for carriage of *Staphylococcus aureus*. None of the controls were screened using the above-mentioned RADT. The cases and controls were, therefore, deemed inappropriate for inclusion in the meta-analysis.

The article published in 2007 by Jensen et al[66] had similar problems as those described above. The inclusion criteria were somewhat unclear. It appears as if primarily patients with a negative outcome of a RADT for GAS were included as cases. Hence, this study was also not included in the meta-analysis.

## Presence of *Fusobacterium necrophorum* in patients with a sore throat

In high or medium quality articles, FN was detected in 18% of cases with a sore throat and a Centor score of 0–4, compared with 7.2% in healthy controls (p<0.00001, $\chi^2$) (table 2). The cumulative positive EPV for FN for the four publications with low or medium risk for bias, including patients with 0–4 Centor scores, was 64% (95% CI 33% to 83%) (figure 3, table 3). In cases with a Centor score of 3–4, FN was detected in 21% (p<0.00001, $\chi^2$) (table 2). The cumulative positive EPV regarding FN for patients with a Centor score of 3–4 was 71% (95% CI 34% to 88%) (figure 4, table 3).

**Table 2** Case–control studies examining *Fusobacterium necrophorum* and group A *Streptococcus* in patients with an acute uncomplicated sore throat in primary care

| Study (Ref) | Design | Method | Age, years (range) | | No of cases and controls | | | % FN detected (n) | | | % GAS detected (n) | | |
|---|---|---|---|---|---|---|---|---|---|---|---|---|---|
| | | | Cases | Controls | Cases | | Controls | Cases | | Controls | Cases | | Controls |
| | | | | | Centor 0–4 | Centor 3–4 | | Centor 0–4 | Centor 3–4 | | Centor 0–4 | Centor 3–4 | |
| Hedin et al[63] | Pro | Culture | 33 (15–48) | 31 (16–46) | 220 | 85 | 128 | 15% (33) | 19% (16) | 3.1% (4) | 30% (66) | 49% (42) | 2.3% (3) |
| Centor et al[62] | Pro | PCR | 22 (15–30) | 24 (15–30) | 312 | 64 | 180 | 21% (64) | 23% (15) | 9.4% (17) | 10% (32) | 16% (10) | 1.1% (2) |
| Kjæruff et al[64] | Pro | Culture | 28 (15–40) | 29 (15–40) | 100 | 29 | 100 | 16% (16) | 21% (6) | 9.0% (9) | 26% (26) | 48% (14) | 3.0% (3) |
| Hayakawa et al 2018[65] | Pro | PCR+culture | 29 (25–37) | 33 (26–36) | 44 | 19 | 31 | 14% (6) | 21% (4) | 6.5% (2) | 11% (5) | 16% (3) | 0.0% (0) |
| *Subtotal (low and medium risk for bias)* | | | | | *676* | *197* | *439* | *18% (119)* | *21% (41)* | *7.2% (32)* | *19% (129)* | *35% (69)* | *1.8 (8)* |
| Jensen et al[66] | Pro | PCR+culture | 25 (18–32) | 22 (18–32) | 105 | – | 92 | 51% (54) | – | 21% (19) | 5.7%* (6*) | – | – |
| Jensen et al 2015[67] | Retro | Culture | 19 (10–40) | 22 (10–40) | 179 | – | 176 | 24% (43) | – | 5.7% (10) | 3.9%* (7*) | – | 0% (0) |
| *Subtotal (high risk for bias)* | | | | | 284 | – | 268 | 34% (97) | – | 11% (29) | 4.6%* (13*) | – | – |
| Total (all six articles) | | | | | 960 | 178 | 707 | 23% (216) | 21% (37) | 8.6% (61) | 15%* (142*) | 39% (69) | 1.1% (8) |

*GAS-tonsillitis was excluded (by general practitioners using rapid antigen tests) in the two articles by Jensen *et al*; thus, those results for GAS were irrelevant for the purpose of this meta-analysis.
FN, *Fusobacterium necrophorum*; GAS, group A *Streptococcus*.

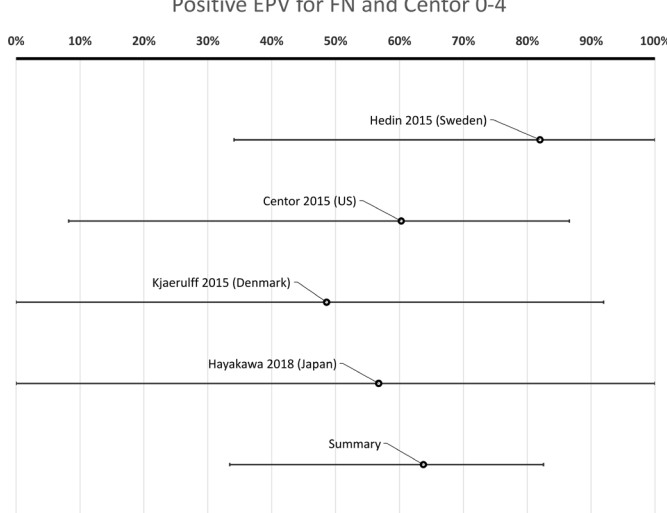

**Figure 3** Probability of a true link between sore throat with Centor score 0–4 and positive *Fusobacterium necrophorum* (FN) test. *Positive etiologic predictive value (EPV) is the probability of a true link between sore throat and FN based on studies with data from both patients and healthy controls.

### *Fusobacterium necrophorum* versus group A *Streptococcus*

When including all cases (Centor score 0–4) in studies with low or medium risk for bias also providing data for GAS in the very same patients, the cumulative positive EPV for a finding of GAS was 93% (95% CI 83% to 99%) (figure 5, table 3). In cases with a Centor score of 3–4, the positive EPV for GAS was 97% (95% CI 91% to 100%) (figure 6, table 3).

### DISCUSSION

This literature review and meta-analysis showed that the P-EPV for FN (detected by culture or PCR) and the uncomplicated acute sore throat was 64% (95% CI 33% to 83%). The corresponding P-EPV for GAS was 93% (95% CI 83% to 99%), based on data from the same publications and patients. When limiting the analyses to the patients with Centor score 3–4, the P-EPV for FN was

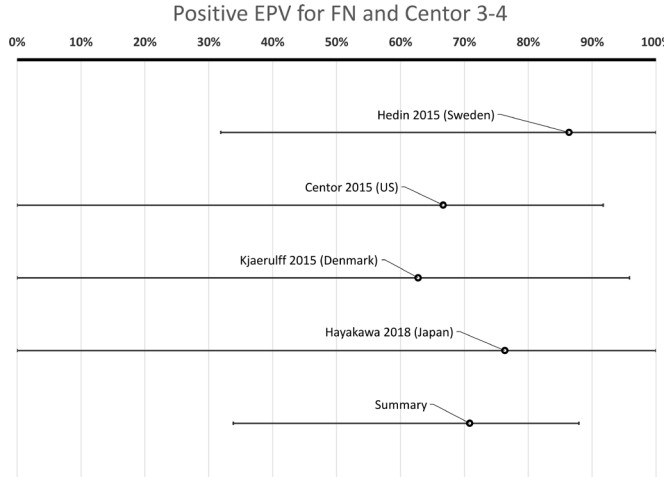

**Figure 4** Probability of a true link between the sore throat with Centor score 3–4 and positive *Fusobacterium necrophorum* (FN) test. *Positive etiologic predictive value (EPV) is the probability of a true link between sore throat and FN based on studies with data from both patients and healthy controls.

71% (95% CI 34% to 88%) and for GAS was 97% (95% CI 91% to 100%).

### Strengths and limitations

A potential limitation is that there were only four available case–control studies presenting the proportion of FN. However, this study is the first systematic literature review with meta-analysis using P-EPV to quantify the clinical relevance of a finding of FN in patients presenting with an acute uncomplicated sore throat in PHC. As such, it represents the current best understanding of the clinical importance of FN in patients with an uncomplicated acute sore throat.

The relatively high prevalence of FN in healthy controls (7.2%) indicates that FN, at least for some patients, is a part of the normal tonsillar flora. The proportion of patients with FN was 18% for Centor score 0–4% and 21% for Centor score 3–4, but the corresponding increase for GAS was from 19% to 35% (table 2). Subsequently, the

| | **Fusobacterium necrophorum** | | **Group A Streptococcus** | |
|---|---|---|---|---|
| **Study** | **Centor 0–4†** | **Centor 3–4†** | **Centor 0–4†** | **Centor 3–4†** |
| Hedin *et al* (Sweden)[63] | 82% (34–100) | 86% (32–100) | 95% (82–100) | 98% (91–100) |
| Centor *et al* (USA)[62] | 60% (8.2–87) | 67% (0.0–92) | 90% (57–100) | 94% (49–100) |
| Kjaerulff *et al* (Denmark)[64] | 49% (0.0–92) | 63% (0.0–96) | 93% (71–100) | 97% (81–100) |
| Hayakawa *et al* (Japan)[65] | 57% (0.0–100) | 76% (0.0–100) | – | – |
| All studies combined‡ | 64% (33–83) | 71% (34–88) | 93% (83–99) | 97% (91–100) |

**Table 3** Positive etiologic predictive value (P-EPV)*

*P-EPV is a method of quantifying the probability of a true link between symptoms and signs (a sore throat) and the finding of a bacterium in the throat while considering the possibility of healthy carriers of the same bacterium. 0% indicates no probability for a true link and 100% indicates a certain link.
†Including either all patients with a sore throat (Centor criteria 0–4) or only those with more prominent symptoms (Centor score 3–4).
‡Only studies with low or medium risk for bias are included.

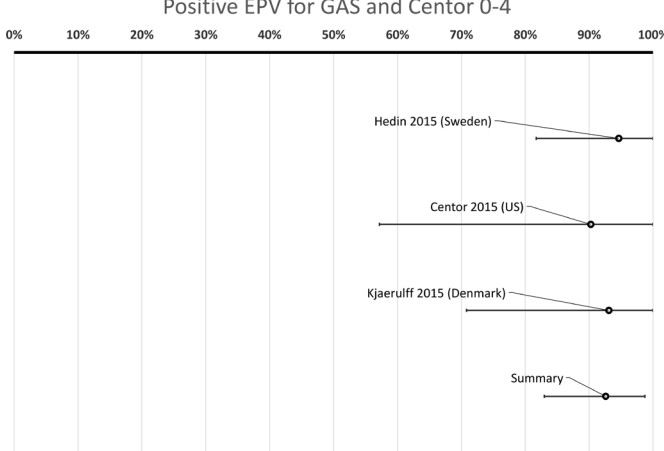

**Figure 5** Probability of a true link between sore throat with Centor score 0–4 and positive group A *Streptococcus* (GAS) test. *Positive etiologic predictive value (P-EPV) is the probability of a true link between sore throat and GAS based on studies with data from both patients and healthy controls. Since the study by Hayakawa *et al* 2018 found no positive tests for GAS among healthy controls, P-EPV could not be calculated for that study separately. However, the results from the four studies selected for meta-analysis are included in the summary.

difference in the 95% CI for P-EPV between patients having 0–4 vs 3–4 Centor scores was very small for FN, even if there was a marginal increase in the point estimate for P-EPV. A larger difference would be expected if FN is the main aetiological agent in a relevant proportion of patients.

The P-EPV numbers indicate that FN plays a role as a pathogen in patients with an uncomplicated acute sore

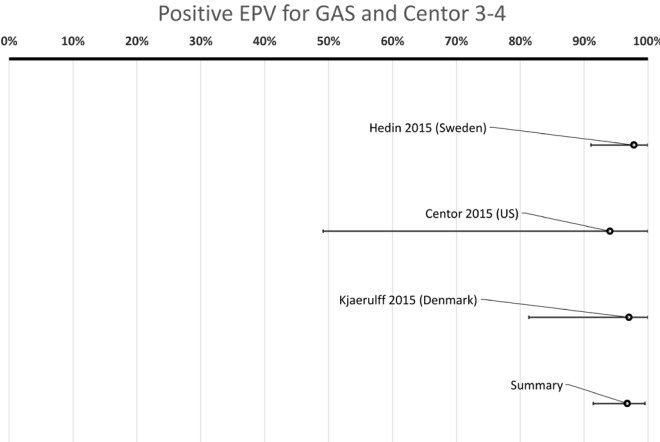

**Figure 6** Probability of a true link between sore throat with Centor score 3–4 and positive group A *Streptococcus* (GAS) test. *Positive etiologic predictive value (P-EPV) is the probability of a true link between sore throat and GAS based on studies with data from both patients and healthy controls. Since the study by Hayakawa *et al* 2018 found no positive tests for GAS among healthy controls, P-EPV could not be calculated for that study separately. However, the results from the four studies selected for meta-analysis are included in the summary.

throat. However, compared with the results for GAS, the association between FN and the uncomplicated acute sore throat appears to be considerably weaker and only marginally higher than the P-EPV of 53% (95% CI 36% to 67%) previously found for adults harbouring Group C streptococci (GCS).[15] Furthermore, the narrow CIs for GAS P-EPV are in contrast with the wide CIs associated with the P-EPV for FN.

The high P-EPV for a finding of GAS in a throat swab is convincing and confirms the already well-established link between GAS and sore throat.[6]

It has been suggested that antibiotic treatment of uncomplicated acute sore throat caused by FN would be cost-effective if it reduces the incidence of Lemierre's syndrome by at least 20%.[26] However, this has not yet been investigated in a clinical trial, and it is unlikely it ever will be due to the very low incidence of Lemierre's syndrome. Other possible reasons for prescribing antibiotic treatment to patients with an uncomplicated acute sore throat caused by FN might be to shorten symptom duration or reduce the incidence of PTA. However, neither one has ever been tested in a clinical trial. Hence, although theoretically possible, we still have no proof that antibiotic treatment is beneficial to patients with an uncomplicated sore throat and presence of FN.

## CONCLUSIONS AND IMPLICATIONS

For uncomplicated acute sore throat in PHC, the CI of P-EPV for FN (33%–83%) is wider and lower compared with the corresponding P-EPV for GAS (83%–99%). The level of certainty for these CIs is deemed as high as it is based on three high quality and one medium quality study including a total of 676 cases and 439 controls. Since the lower limit for the 95% CI for FN is well above 0%, we can, with a high level of certainty, state there is an association between FN and the uncomplicated acute sore throat in PHC. However, it is weaker than the same association for GAS.

We are not aware of any studies showing that antibiotic treatment has beneficial effects on the duration or severity of symptoms in an FN-associated acute sore throat. Nor do we have any evidence that antibiotic treatment to patients with an uncomplicated acute sore throat reduces the incidence of the life-threatening Lemierre's syndrome. Hence, in the absence of this evidence, we do not recommend routinely searching for FN in throat swabs or prescribing antibiotics to patients with a GAS negative uncomplicated acute sore throat in PHC. However, to our knowledge, at least one randomised controlled trial focusing on GAS-negative patients with a sore throat, and analysing the presence of FN, is underway. Hence, our current advice may in the future have to be revised.

More future studies should focus on randomising patients with an uncomplicated acute sore throat and presence of only FN to treatment with antibiotics or placebo in order to assess whether the treatment is effective to reduce duration and intensity of symptoms, and,

more importantly, if complications such as PTA can be prevented. The prevalence of Lemierre's syndrome is so low that any effect of antibiotics on its prevalence most likely would need to be estimated using other designs than a simple clinical trial comparing antibiotics with placebo.

**Author affiliations**
[1]General Practice / Family Medicine, School of Public Health and Community Medicine, Institute of Medicine, Sahlgrenska Academy, University of Gothenburg, Gothenburg, Sweden
[2]Research, Education, Development & Innovation, Primary Health Care, Region Västra Götaland, Borås, Sweden
[3]Centre for Antibiotic Resistance Research (CARe), University of Gothenburg, Gothenburg, Sweden
[4]Faculty of Medicine and Health Sciences, Linköping University, Linköping, Sweden
[5]Futurum Academy for Health and Care, Region Jönköpings County, Jönköping, Sweden
[6]Department of Clinical Sciences in Malmö, Family Medicine, Lund University, Malmö, Sweden

**Acknowledgements** The authors would like to thank Professor Robert Centor who willingly provided further information from his study "The clinical presentation of Fusobacterium-positive and streptococcal-positive pharyngitis in a university health clinic: a cross-sectional study",[62] enabling us to calculate separate P-EPV for patients with 0–4 and 3–4 Centor criteria.

**Contributors** RG was responsible for the conception of the idea. The study's overall design was made by RG, SM, PDS, SP and KH. SM and SP were responsible for assessing the eligibility and extract data from publications. Statistical analysis, interpretation of results and writing of the manuscript were made by all authors (RG, SM, PDS, SP and KH). All authors (RG, SM, PDS, SP and KH) approved the final version of the manuscript.

**Funding** Funding for this project came from the Local Research and Development Council, Södra Älvsborg, Sweden (reference number VGFOUSA-937533). Funding was also received from Research and Development Centre for Primary Healthcare, Borås, Sweden (reference number VGFOUSA-P-872381).

**Competing interests** None declared.

**Patient consent for publication** Not required.

**Provenance and peer review** Not commissioned; externally peer reviewed.

**Data availability statement** Data are available upon reasonable request. The raw data files are available from the corresponding author upon request.

**ORCID iD**
Stefan Malmberg http://orcid.org/0000-0002-8094-2086

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
