## [Reviewer comments · BMJ Open]

ARTICLE DETAILS

TITLE (PROVISIONAL)	Acute Sore Throat and Fusobacterium Necrophorum in Primary Health Care – A Systematic Review and Meta-analysis
AUTHORS	Malmberg, Stefan; Petrén, Susanna; Gunnarsson, Ronny; Hedin, Katarina; Sundvall, Pär-Daniel

VERSION 1 – REVIEW

REVIEWER	Klug, Tejs Ehlers Aarhus Universitet
REVIEW RETURNED	29-Aug-2020

GENERAL COMMENTS	This systematic review deals with the probability for an association between uncomplicated acute sore throat and Fusobacterium necrophorum (FN). The major addition to the current literature is the calculation of P-EPV for FN and Group A streptococci (GAS). The review is well conducted according to the PRISMA guidelines and well written. I have no major concerns regarding the study or manuscript. The P-EPV seems to be a clinically relevant and useful parameter, which is an important contribution to the understanding of FN in patients with sore throat. I only miss one elaboration: How do you interpret P-EPV? – on an individual level or on an overall level? I.e. P-EPV regarding FN (Centor score 0-4) was 64%; does that mean that the probability of FN being the true pathogen is 64% in a patient with the finding of FN in a throat swab or does it mean the probability of a true association between FN and acute sore throat in general is 64%. Please explain this in the manuscript. And maybe readers need more elaboration on the matter: what does P-EPV = 50% mean? What would P-EPV be, if a bacterium was found equally often in patients and controls (0%?)?
---

REVIEWER	Ebell, Mark University of Georgia, Epidemiology and Biostatistics
	I have collaborated on one study with one of the co-authors, Dr. Gunnarson
REVIEW RETURNED	29-Aug-2020

GENERAL COMMENTS	General comments This is an important contribution to the literature and takes a novel approach to analysis of sore throat data to determine causation. More clarity around calculation of P-EPV and meta-analysis methods, and either enhancing the forest plots or reporting data in a tabular fashion would also be important.
---

	Introduction Page 4, Line 14: Consider adding "...and young adults." Methods Page 5, line 4: More appropriate to say it was registered with PROSPERO rather than published, I think. Page 7, line 11: Was "high methodologic quality" defined before the study? Or post hoc? How was it defined? Also, the abstract says "4 studies of medium to high quality", which is a bit different. Please reconcile these differences. Page 7, lines 18-30: Please provide more information about how P-EPV is calculated. It was new to me, and probably will be new to other readers as well. Page 7, lines 11-16: Please provide more detail about the meta-analysis as well. Did you use a random effects model to calculate summary estimates of the P-EPV, which is (I think) a proportion? Did you use a measure of heterogeneity such as I²? Which statistical software was used? Results Page 7, lines 48-57: You mention reviewing the reference lists for additional articles in the Methods but I don't see where you did that. Quality assessment Figure 2: How are low, intermediate and high risk of bias defined for each category? When using QUADAS-2 that is reported explicitly. Page 10, line 32: How is EPV calculated? I thought perhaps (18-7)/18 but that is 61%. This is why it would be helpful to tell us how it is calculated in the methods. Do not make readers dig for reference 15. Page 10, line 43/44: The heading is inaccurate, as there is no statistical comparison being made. Figures 3-6: This is a case where I think providing point estimates and confidence intervals in a table would be more useful and precise than doing so graphically. You could have separate tables for FN and GAS, and move the figures to the appendix. Or create forest plots that include the raw data and confidence intervals numerically, as we are used to seeing with Stata or R in other meta-analyses and forest plots, along with a measure of heterogeneity such as I². Discussion Page 11, lines 34-36: You can use random effects meta-analysis of proportions, it does not only provide odds ratios. Or am I misunderstanding? Also, P-EPV is not a technique for meta-analysis, is it? It is the statistic you calculate to represent the proportion of cases caused by FN and GAS when it is detected, and you then do meta-analysis to calculate the summary estimate for the 4 studies combined. Correct? Page 11, line 59: Tweaking? How would you "tweak" this? Page 12, lines 27-34: Do you have P-EPV estimate for other pathogens from other previously published studies, so we can have more context around this 64% estimate. Page 13, lines 14-16: Hard to judge how strong or weak 64% is without more context. It seems to me that if it is the etiologic agent 64% of the time it is detected in someone with sore throat, then that is relatively high (although lower than GAS, of course).
--	--

REVIEWER	Brignardello-Petersen, Romina McMaster University
REVIEW RETURNED	30-Oct-2020

GENERAL COMMENTS	This systematic review has many methodological and reporting limitations, which makes my doubt the process. I summarize the most important concerns below. Only reading the abstract I am confused. It is not clear to me how the objective of assessing the association between FN and sore throat is best answered by the statistical analysis chosen by the authors. It is also unclear why they chose to use the reference group chosen. Based on the "article summary", seems like the purpose in the abstract should be different The abstract has no description of eligibility criteria for the included studies, which makes the issue above more problematic. The authors should clarify the type of study design they included. Seems like the type of studies they searched for were cross-sectional, given the nature of how the data should have been collected in the studies. The fact that there was a healthy group as a comparator does not make a study a case control study. Authors make it seem like they don't understand research methods when they say in lines 27-30 that they included prospective case control studies, which are not an existing study design. Issues like this decrease the credibility of this study. Can I infer that the authors mean "title and abstract screening" when they say "first screening" Or does it mean only "title screening"? It is concerning that only one author conducted this step, and it is not clear what was done by one versus 2 authors. Given that the authors truly included cross sectional studies, the tool they used for assessing methodological quality does not seem appropriate It is unclear why the authors chose the analysis they did, instead of using a simple OR to qualify the association they were exploring. There is no justification provided in the methods. It is also unclear why, all the sudden there is another comparison (between FN and GAS), which is not part of the objectives of the review, nor eligibility criteria. How can we know that the authors found and included all relevant evidence for this extra piece of information? There is no assessment of certainty of the evidence.
---

VERSION 1 – AUTHOR RESPONSE

===== Reviewer 1 Tejs - Ehlers Klug =====

This systematic review deals with the probability for an association between uncomplicated acute sore throat and *Fusobacterium necrophorum* (FN). The major addition to the current literature is the calculation of P-EPV for FN and Group A streptococci (GAS). The review is well conducted according to the PRISMA guidelines and well written. I have no major concerns regarding the study or manuscript. The P-EPV seems to be a clinically relevant and useful parameter, which is an important contribution to the understanding of FN in patients with sore throat.

Author response: We are happy that the reviewer overall appreciated our manuscript with no major concerns.

I only miss one elaboration: How do you interpret P-EPV? – on an individual level or on an overall level? I.e. P-EPV regarding FN (Centor score 0-4) was 64%; does that mean that the probability of FN being the true pathogen is 64% in a patient with the finding of FN in a throat swab or does it mean the probability of a true association between FN and acute sore throat in general is 64%. Please explain this in the manuscript. And maybe readers need more elaboration on the matter: what does P-EPV = 50% mean? What would P-EPV be, if a bacterium was found equally often in patients and controls (0%?)?

*Author response: **We agree** with the reviewer that some assistance on how to interpret P-EPV is required and we have added that to the manuscript. Briefly speaking if a bacterium is found equally often in patients and controls the point estimate of P-EPV will be 0% with a 95% confidence interval from 0.0% and the upper limit will be determined by the sample size. P-EPV is strictly speaking the probability for a true connection between the bacterial finding and the specified illness in the data set investigated while considering that some patients are merely carriers ill from something else. As in all research the reader needs to decide to what extent the investigated data set resembles their own patients and this determines the generalisability.*

===== **Reviewer 2 - Mark Ebell** =====

This is an important contribution to the literature and takes a novel approach to analysis of sore throat data to determine causation. More clarity around calculation of P-EPV and meta-analysis methods, and either enhancing the forest plots or reporting data in a tabular fashion would also be important.

*Author response: **We agree** that since P-EPV is a new novel statistical approach a more thorough explanation of how P-EPV is calculated and how to interpret it is required. This has been added to the manuscript.*

Introduction Page 4, Line 14: Consider adding "...and young adults."

*Author response: **We agree** and added "...and young adults".*

Methods

Page 5, line 4: More appropriate to say it was registered with PROSPERO rather than published, I think.

*Author response: **We agree** and changed accordingly.*

>Page 7, line 11: Was "high methodologic quality" defined before the study? Or post hoc? How was it defined? Also, the abstract says "4 studies of medium to high quality", which is a bit different. Please reconcile these differences.

*Author response: We defined high methodological quality in advance using the same definition as in the meta-analysis of group C Streptococci published in 2020 by Gunnarsson et al. **We agree** that this was not expressed in the manuscript. We have added clarity about this in the revised manuscript and also added new table 1.*

Page 7, lines 18-30: Please provide more information about how P-EPV is calculated. It was new to me, and probably will be new to other readers as well.

Author response: **We agree** that since P-EPV is a new novel statistical approach a more thorough explanation of how P-EPV is calculated and how to interpret it is required. This has been added to the manuscript both in the methods section and also in new Appendix 2.

Page 7, lines 11-16: Please provide more detail about the meta-analysis as well. Did you use a random effects model to calculate summary estimates of the P-EPV, which is (I think) a proportion? Did you use a measure of heterogeneity such as I²? Which statistical software was used?

Author response: **We agree** and added three new paragraphs in the methods section (some of these moved from the discussion). We think these changes answers the reviewer's questions. The software used was an Excel file tested against two separate online EPV calculators (<https://infovoice.se/fou/epv/calc/index.htm> and <https://science-network.tv/epv-calculator/>).

Results

Page 7, lines 48-57: You mention reviewing the reference lists for additional articles in the Methods but I don't see where you did that.

Author response: Reviewing reference lists did not reveal any more relevant publication. **We agree** with the reviewer that this should be explained better and we have adjusted this in the revised version of the manuscript.

Quality assessment Figure 2: How are low, intermediate and high risk of bias defined for each category? When using QUADAS-2 that is reported explicitly.

Author response: This was defined in advance but **we agree** this is not expressed clearly in the manuscript. We appreciate the reviewer pointed us to this and we made changes accordingly also adding new table 1.

Page 10, line 32: How is EPV calculated? I thought perhaps (18-7)/18 but that is 61%. This is why it would be helpful to tell us how it is calculated in the methods. Do not make readers dig for reference 15.

Author response: **We agree** that since P-EPV is a new novel statistical approach a more thorough explanation of how P-EPV is calculated and how to interpret it is required. This has been added to the manuscript and we also added appendix 2 providing the formula in detail.

Page 10, line 43/44: The heading is inaccurate, as there is no statistical comparison being made.

Author response: The comparison between FN and GAS is not a statistical comparison where a p-value is calculated. **We agree** with the reviewer that this might be misleading. Hence, we changed "comparison" to "versus".

Figures 3-6: This is a case where I think providing point estimates and confidence intervals in a table would be more useful and precise than doing so graphically. You could have separate tables for FN and GAS, and move the figures to the appendix. Or create forest plots that include the raw data and confidence intervals numerically, as we are used to seeing with Stata or R in other meta-analyses and forest plots, along with a measure of heterogeneity such as I².

Author response: **We agree** with the reviewer that some readers prefer exact numbers in a table while others prefer the graphs. Hence, we added new table 3 stating exact point estimates and confidence intervals for P-EPV. A measure of heterogeneity is not given since this is not a random effects model and the reasons for this is now explained in the revised methods section. We kept figure 3-6 as they are likely to appeal to many readers as well.

Discussion

Page 11, lines 34-36: You can use random effects meta-analysis of proportions, it does not only provide odds ratios. Or am I misunderstanding? Also, P-EPV is not a technique for meta-analysis, is it? It is the statistic you calculate to represent the proportion of cases caused by FN and GAS when it is detected, and you then do meta-analysis to calculate the summary estimate for the 4 studies combined. Correct?

Author response: A true random effect model would provide odds ratios as well as an estimate of heterogeneity by calculating I^2 . P-EPV used as a method for meta-analysis does not provide I^2 statistics. However, P-EPV has other advantages as described in the revised methods section. Furthermore, the I^2 statistics is very unreliable when the number of included publications is small (<https://doi.org/10.1186/s12874-015-0024-z>). We knew already from the start that the number of publications available would be small and we finally ended up with only four studies. In this situation we found P-EPV to be more useful for this particular review.

Page 11, line 59: Tweaking? How would you “tweak” this?

*Author response. **We agree** with the reviewer that “tweak” is a vague expression. We replaced it with “adjust”.*

Page 12, lines 27-34: Do you have P-EPV estimate for other pathogens from other previously published studies, so we can have more context around this 64% estimate.

*Author response: **We agree** this leaves the reader in a limbo. There are previous publications using P-EPV. We added a section in the discussion where we cite one and put the 64% estimate in perspective.*

Page 13, lines 14-16: Hard to judge how strong or weak 64% is without more context. It seems to me that if it is the etiologic agent 64% of the time it is detected in someone with sore throat, then that is relatively high (although lower than GAS, of course).

*Author response: **We agree** that 64% can't directly be translated to strong or weak. However, it can be compared to P-EPV with GAS or group C Streptococci. The important question is if patients with an acute uncomplicated sore throat should be prescribed antibiotics rests on the scientific evidence we have stating that antibiotics actually puts the patient in a better position. This evidence exists for patients harbouring GAS but is currently lacking for patients with presence of FN. Fortunately, a study for FN are underway that will answer this question and this study may well change our view on this in the future. We rewrote sections of the discussion to better reflect this.*

===== **Reviewer 3 - Romina Brignardello-Petersen** =====

This systematic review has many methodological and reporting limitations, which makes my doubt the process. I summarize the most important concerns below.

Only reading the abstract I am confused. It is not clear to me how the objective of assessing the association between FN and sore throat is best answered by the statistical analysis chosen by the authors. It is also unclear why they chose to use the reference group chosen. Based on the "article summary", seems like the purpose in the abstract should be different

*Author response: **We agree** that this could have been expressed more clearly. The rewritten methods section much better explains the rationale for the chosen statistical analysis. It was not possible to fit this explanation within the abstract so it is found in the methods section of the manuscript.*

The abstract has no description of eligibility criteria for the included studies, which makes the issue above more problematic.

*Author response: **We agree** that inclusion criteria should be stated in the abstract. Hence, the inclusion criteria have been added to the abstract.*

The authors should clarify the type of study design they included. Seems like the type of studies they searched for were cross-sectional, given the nature of how the data should have been collected in the studies. The fact that there was a healthy group as a comparator does not make a study a case control study. Authors make it seem like they don't understand research methods when they say in lines 27-30 that they included prospective case control studies, which are not an existing study design. Issues like this decrease the credibility of this study.

*Author response: Let us clarify that the authors have a reasonable understanding of research methods with 17 peer-reviewed systematic literature reviews published so far (and several more in pipeline). **We understand** what the reviewer means and believe the concerns raised is due to an unfortunate use of words causing confusion. The exposure in this case is presence of a bacteria and the sore throat is the disease. In this situation the time elapsed between exposure begins and presence of the disease is not decades (as in many traditional case-control studies) but rather a couple of days. The exposure remains within the patient until they start to get well. Hence, although a traditional case-control study is longitudinal we can study the exposure in cases and controls while at the same time classifying them as cases and controls. This makes it also being a cross-sectional study. Hence, the types of studies we include can both be classified as case-control studies and as cross-sectional studies. The choice of classification does not influence the research question, choice of statistical procedure or the interpretation of the result. We still believe case-control studies is a better label making the reader interested in this particular topic understand better.*

*Traditional case-control studies are retrospective in their nature. Some studies found used retrospective data while others planned a research project and after ethics approval prospectively collected samples for bacterial analysis from patients as well as healthy controls. We labelled the latter prospective case-controls but after reading the reviewers comment **we agree** this can be perceived as confusing. We have removed "prospective and retrospective" from the manuscript to avoid confusion.*

Can I infer that the authors mean "title and abstract screening" when they say "first screening" Or does it mean only "title screening"?. It is concerning that only one author conducted this step, and it is not clear what was done by one versus 2 authors.

*Author response: **We agree** with the reviewer that simply stating "screened" is not precise enough. Many publications initially identified by the search were animal research obviously outside the scope of this review and they were removed by one author after screening of title and abstract. The remaining publications were again screened by title and abstract by two authors. They started screening sitting together in the same room discussing each publication to ensure they aligned their judgement. They then continued screening separately but had a joint discussion whenever they decided differently if a publication should be kept or removed. This process has been further clarified in the manuscript.*

Given that the authors truly included cross sectional studies, the tool they used for assessing methodological quality does not seem appropriate

Author response: The important aspects of methodological quality for this particular review is given in new table 1 and figure 2. Hence, we removed the reference to the SIGN 50 checklist. We also added a detailed description of what constitutes overall high or low methodological quality.

It is unclear why the authors chose the analysis they did, instead of using a simple OR to qualify the association they were exploring. There is no justification provided in the methods.

*Author response: **We agree** that this was not expressed clearly enough. A true random effect model would provide odds ratios as well as an estimate of heterogeneity by calculating I^2 . P-EPV used as a method for meta-analysis does not provide I^2 statistics. However, P-EPV has other advantages as described in the now revised and extended methods section.*

It is also unclear why, all the sudden there is another comparison (between FN and GAS), which is not part of the objectives of the review, nor eligibility criteria. How can we know that the authors found and included all relevant evidence for this extra piece of information?

Author response: P-EPV was found to be 64%. To put this in perspective we compared this with the same P-EPV obtained when analysing cases and controls for GAS. GAS is the most well-known bacteria causing a sore throat and all readers interested in the sore throat will know that. We deliberately did not search for all studies comparing the prevalence of GAS between patients with a sore throat (cases) and controls. We only included the same publications presenting this data for FN. The advantage of doing so is that the comparison of the P-EPV between FN and GAS will be done on exactly the same patients with samples taken at the very same occasion making them more comparable. We agree with the reviewer that the comparison of P-EPV between FN and GAS should be mentioned in the aim so we added it there.

There is no assessment of certainty of the evidence.

*Author response: **We agree** with the reviewer that providing certainty of evidence would improve the manuscript. Hence, we added this in the conclusions.*

VERSION 2 – REVIEW

REVIEWER	Ebell, Mark University of Georgia, Epidemiology and Biostatistics I have previously written a meta-analysis of the prevalence of fusobacterum necrophorum, and have collaborated in the past on one study with Dr. Gunnarson
REVIEW RETURNED	01-Feb-2021

GENERAL COMMENTS	page 6, line 35: Would say "Moderate risk of bias" rather than "Medium quality" to be consistent with the other categorizations. It is also not clear what lies between 0 and 1 categories at high risk of bias, i.e. how does one get into that medium category? Appendix 2: I'm not a statistician, so I may be misunderstanding the formula. But it seems to me that in the formula, the denominator should be patients without sore throat (i.e. asymptomatic controls) and therefore S-, not S+. Otherwise, the authors have responded appropriately and I believe their revised manuscript is deserving of publication.
--

REVIEWER	Brignardello-Petersen, Romina McMaster University
REVIEW RETURNED	10-Feb-2021

GENERAL COMMENTS	I thank that authors for clarifying and editing the manuscript to address my concerns. I still, however, have 2 important concerns that have not been addressed satisfactorily. 1. Choice of method of analysis and results presentation: the rationale for choosing the PEPV is still not sufficiently justified in my opinion. First, many may argue that clinicians are much more used to reading articles with ORs rather than PEPV, thus I don't think that it's true that the OR would be less understood by clinicians. Second, in their rebuttal letter the authors mention issues around random
--

	effects meta-analysis that are appropriate, but it is not clear how their meta-analyses fixed this problem (they just mention they did a meta-analysis of PEPV, without specific statistical methods for such meta-analysis described (i.e., The authors highlight issues around random effect meta-analysis, but it is not clear how their own meta-analysis dealt with those, since these issues are not particular to the measure of association- OR vs PEPV- but to meta-analyses in general)). 2. Assessment of the certainty of the evidence: It is now widely known that it is important to assess the estimates of effects/ association/ whatever summary measure you use in your meta-analysis. That is, systematic reviewers don't stop at providing the summary estimate, but also assess how much confidence (certainty, trust), they have on the summary estimate. This assessment is a formal step of SRs, and its results are directly related to the conclusions authors can draw. The assessment considers issues around risk of bias, inconsistency across studies, imprecision, indirectness/ applicability, and publication bias. Presenting the estimate and its 95% CI as a measure of statistical uncertainty is not the same as assessing the certainty of the evidence. - Minor point- the article summary could be more helpful. Although it is helpful to learn that the PEPV can take values from 0-100, it would be more helpful to mention what it actually was in this SR.
--	--

VERSION 2 – AUTHOR RESPONSE

===== Reviewer 2 - Mark Ebell =====

page 6, line 35: Would say "Moderate risk of bias" rather than "Medium quality" to be consistent with the other categorizations. It is also not clear what lies between 0 and 1 categories at high risk of bias, i.e. how does one get into that medium category?

Author response: We agree and are sorry we missed to fix this in the previous revision. It has now been changed to "moderate risk of bias". We first identify publications of low and high risk of bias. The rest are defined as having a moderate risk of bias: "Overall high quality was defined as having a low risk of bias in all criteria. Having a high risk of bias in any criteria made the study to be of overall low quality. The rest was classified as having an overall moderate risk of bias."

Appendix 2: I'm not a statistician, so I may be misunderstanding the formula. But it seems to me that in the formula, the denominator should be patients without sore throat (i.e. asymptomatic controls) and therefore S-, not S+.

Author response: We assume the reviewer refers to the formulae for P-EPV in appendix 2. $P(T+|S+D-)$ refers to individuals having a positive throat swab (T+), having a sore throat (S+) but the sore throat is caused by something else (D-). (D-) refers to that these individuals do not have a sore throat caused by FN/GAS which is our definition of (D+). The formulae in appendix 2 is correct and we refer to the original publication from 2002 (reference 14) where professor Jan Lanke is co-author (a prominent biostatistician in Sweden).

Otherwise, the authors have responded appropriately and I believe their revised manuscript is deserving of publication.

Author response: We are happy to understand the reviewer is happy with the manuscript.

===== Reviewer 3 - Romina Brignardello-Petersen =====

Choice of method of analysis and results presentation: the rationale for choosing the PEPV is still not sufficiently justified in my opinion. First, many may argue that clinicians are much more used to reading articles with ORs rather than PEPV, thus I don't think that it's true that the OR would be less understood by clinicians. Second, in their rebuttal letter the authors mention issues around random effects meta-analysis that are appropriate, but it is not clear how their meta-analyses fixed this problem (they just mention they did a meta-analysis of PEPV, without specific statistical methods for such meta-analysis described (i.e., The authors highlight issues around random effect meta-analysis, but it is not clear how their own meta-analysis dealt with those, since these issues are not particular to the measure of association- OR vs PEPV- but to meta-analyses in general)).

*Author response: **We agree** that clinicians involved in research are much more likely to be familiar with odds ratios than P-EPV. However, most clinicians, at least in primary health care where most of these patients are managed, are not engaged in research and they are unfamiliar with both odds ratios and P-EPV. For clinicians unfamiliar with research we believe P-EPV is much easier to understand than odds ratios. Clinicians managing these patients will find P-EPV easy to understand even if they are familiar with odds ratios.*

P-EPV does not present a numeric value for the between-study variation. The I² statistics in traditional random effect models is uncertain in this case with few publications. In the situation with few publications it is our opinion that figure 3-6 provides a graphical presentation of the between-studies variation. We made some minor changes to the methods section for clarification.

Assessment of the certainty of the evidence: It is now widely known that it is important to assess the estimates of effects/ association/ whatever summary measure you use in your meta-analysis. That is, systematic reviewers don't stop at providing the summary estimate, but also assess how much confidence (certainty, trust), they have on the summary estimate. This assessment is a formal step of SRs, and its results are directly related to the conclusions authors can draw. The assessment considers issues around risk of bias, inconsistency across studies, imprecision, indirectness/ applicability, and publication bias. Presenting the estimate and its 95% CI as a measure of statistical uncertainty is not the same as assessing the certainty of the evidence.

Author response: This type of review is new and no existing method to grade the certainty of evidence we found apply. There are recommendations when reviewing evaluations of diagnostic tests (which does not directly apply to our review) where the 95% confidence interval for tests characteristics such as sensitivity, specificity etc can be considered as a kind of "certainty rating" (<https://www.sciencedirect.com/science/article/pii/S0895435618310692>). Predictive values are commonly used in test evaluations together with sensitivity and specificity. This is the closest we have come to a certainty rating that may apply to the particular scenario studied in this systematic review. We also translate this in the conclusion to "...we can with a high level of certainty state there is an association between FN and the uncomplicated acute sore throat in PHC"

Minor point- the article summary could be more helpful. Although it is helpful to learn that the PEPV can take values from 0-100, it would be more helpful to mention what it actually was in this SR.

Author response: We agree and added this to the article summary.